# Instrumental and Sensory Characteristics of Commercial Korean Rice Noodles

**DOI:** 10.3390/foods10112885

**Published:** 2021-11-22

**Authors:** Ah-Hyun Lee, Seon-Min Oh, Sang-Jin Ye, Hui-Yun Kim, Ji-Eun Bae, Jong-Hyun Choi, Byung-Yong Kim, Moo-Yeol Baik

**Affiliations:** Department of Food Science and Biotechnology, Institute of Life Science and Resources, Graduate School of Biotechnology, Kyung Hee University, Yongin 17104, Korea; ahyun042@naver.com (A.-H.L.); seonminoh@khu.ac.kr (S.-M.O.); lifesci2015@naver.com (S.-J.Y.); mikoel@khu.ac.kr (H.-Y.K.); wise123@khu.ac.kr (J.-E.B.); choi6167@khu.ac.kr (J.-H.C.); bykim@khu.ac.kr (B.-Y.K.)

**Keywords:** rice noodles, bending test, stress–relaxation, failure stress, modulus of elasticity, sensory test

## Abstract

In this study, the rheological properties of several commercial rice noodle strands were investigated. In the bending test, failure stress decreased as the cooking temperature increased from 80 to 90 °C, and the cooking time increased from 3 to 4 min for higher rice content noodles (>60%). The stress–relaxation test and sensory tests were carried out with bundles of noodles to investigate correlations with the bending test. The modulus of elasticity was higher at 80 than 90 °C. However, no correlation was found between cooking temperature and the rheological properties of lower rice content noodles. In the stress relaxation test, the deviation was larger due to the empty space in the bundle. In the correlation analysis, sensory stickiness was correlated with a modulus of elasticity in the bending test. Comparing the bending and stress–relaxation tests, each instrumental variable showed differences in the rheological properties of rice noodles in strands and bundles. However, the bending test measured with noodle strands seemed to be most suitable as a method of measuring the rheological properties of rice noodles.

## 1. Introduction

Rice has been cultivated and consumed extensively in Asia, and its characteristics differ depending on amylose content and grain type. Typical rice produced in Korea is short-grain japonica rice and is used as a staple food due to the fact of its high digestibility and stability; gluten-free products based on rice have been developed in recent years [1,2]. In Korea, the rice noodle is a representative rice-based product and prepared by two methods depending on the processing properties: extruded and sheeted noodles. [3]. The extrusion method is suitable for rice noodles in Korea, and more than 80% of Korean commercial rice noodles are produced via an automatic system [4].

Numerous studies have reported on the properties of noodles, and various factors, such as amylose content [5,6], hydrothermal modification of flour [7], and hydrocolloids [8,9], affect the quality of noodles. Research to evaluate the quality of noodles has been carried out and can be categorized into mechanical tests and sensory evaluations. Since starting as a simple taste test, sensory evaluation has consequently been developed to make objective and stable measurements, and it has been highly developed through data collection and statistical analysis. Currently, sensory evaluation combined with other fields, such as genomics, brain imaging, and instrumental analysis, has advanced into a sophisticated decision-making tool [10]. In a sensory test, it is necessary to consider many factors such as the limitation of the sample size, panel training, and reproducibility of the evaluation [11].

To overcome these problems, several mechanical tests have been developed that utilize time- and cost-effective instruments. For mechanical testing of rice noodles, many parameters, such as compression, cutting, tensile, and viscoelastic behavior, are investigated using uniaxial testing [12]. Ross [13] suggested 5–10 replicates of each sample to obtain valid experimental data for reasons such as inhomogeneity and microcracks of cooked noodles. To determine the viscoelastic properties of noodles, stress relaxation has been applied as a principal means [14], and this property can be obtained through the stress relaxation test using Maxwell or Kelvin models, which consist of an elastic spring and a viscous dashpot [15]. Shiau et al. [16] confirmed that the three-point Maxwell and Peleg-Normand models were suitable for measurement of the stress–relaxation of dietary fiber-rich noodles. Meanwhile, the bending test of food products was applied to investigate microstructural and molecular mechanisms and related Young’s modulus and fracture toughness [17]. It has been suggested that fracture toughness, which is the geometry of failure for rod-like food pieces in the mouth, can be better demonstrated using three-point bend loading [18]. According to Jeong et al. [19], a greater force was needed to break noodles with high amylose rice flour via a three-point bending test, and there was correlation between breaking stress and pasting parameters.

Several studies have suggested a correlation between the mechanical and sensory properties of noodles [20,21]. Choi, Cho, and Koh [2] indicated that, although there is some correlation between factors that can predict the quality of rice noodles and their actual quality, it is difficult to obtain a high correlation. Since Korean rice noodles produced mostly by extrusion are thin (1.3–2.0 mm) and uneven, it is necessary to determine the appropriate instrumental measurements. However, the technological properties of noodles have commonly been measured in bundles rather than strands of noodles. Therefore, this study was designed to investigate the mechanical properties of commercial rice noodle strands using the bending test compared with the stress–relaxation of bundles of noodles. In addition, we investigated the correlations between mechanical properties and sensory evaluation.

## 2. Materials and Methods

### 2.1. Materials

Nine kinds of commercial rice noodles in the Korean market were used for analysis, and the noodles’ appearance and content of rice flour are shown in Figure 1. Rice noodles are abbreviated as “R”, and samples were numbered from R1 to R9. The detailed ingredients of the rice noodle samples are shown in Table 1.

### 2.2. Bending Test

Ten noodle strands were cooked at 80 and 90 °C for 3–4 min using a distilled water and cooled to room temperature using tap water for 30 s. After draining, a three-point bending test of the rice noodles was conducted using a rheometer (Sun Co. CR-150, Tokyo, Japan) with a wire cutter probe. The load cell was set at 0.98–1.96 N, and the crosshead and chart speeds were 300 mm/min. Failure stress (σ) and modulus of elasticity (*E*) of the noodle strands were obtained using the following equations [22] (Figure 1B).
(1)$$Failure\;stress\;\left(\sigma \right)=\;\frac{F\cdot l}{\pi \cdot R^{4}}$$
(2)$$Modulus\;of\;elasticity\;\left(E\right)=\frac{P\cdot l}{48\cdot I\cdot \delta }$$
where *F* is the loading force, *l* is the distance between supports, *I* is the moment of inertia (π·*R*^4^/4), *R* is the radius of the noodle, and the maximum deflection (δ) is half the distance between supports (l/2).

### 2.3. Stress–Relaxation Test

For the stress–relaxation test, rice noodles (30 g) were cooked at 90 °C for 4 min using a distilled water and cooled to room temperature using a tap water for 30 s. After draining, the noodles were deposited to a 3 cm height in a 100 mL beaker. The stress–relaxation of cooked rice noodles were assessed using a rheometer with a parallel plate probe. Tests were carried out at a 0.15 strain rate with 300 mm/min crosshead speed, and the samples were maintained in the deformed state for 1 min. Instantaneous and equilibrium stresses were recorded (Figure 1C). The data were applied to a three-element Maxwell model composed of two elastic springs on a viscous dashpot, and then these values were used in the following equations [23].
(3)$$E_{1}=\sigma_{0}\cdot \varepsilon_{0}$$
(4)$$\;\frac{\sigma_{e}}{\varepsilon_{0}}=\frac{E_{1}\cdot E_{3}}{\left(E_{1}+E_{3}\right)}\;\;\;\;$$
(5)$$\tau_{b}=\frac{\eta_{3}}{\left(E_{1}+E_{3}\right)}$$
where σ_0_ is instantaneous stress, σ_e_ is equilibrium stress, ε_0_ is initial strain, τ_b_ is relaxation time, and *η*_3_ is the viscous constant. *E*_1_ and *E*_2_ indicate the elastic element constant and parallel elastic element constant, respectively.

### 2.4. Sensory Evaluation

The hardness and stickiness of cooked noodles at 90 °C for 4 min were evaluated by 15 trained panelists using a seven-point scale (Figure 2). The panelists were selected as follows: (1) people with normal taste evaluation and expression skills; (2) people who were not allergic to certain foods; (3) people who were not color- and taste-blind. In addition, the following training was performed before sensory evaluation: (1) explanation of the sensory evaluation method; (2) tasting method (spoon, cup, drink, etc.); (3) a specific description of the questions, terms, and scale in the evaluation form. Hardness was judged as the force required to bite the noodle strand. Stickiness was defined as the adherence of the noodle surface to the mouth. At the same time, overall preference was evaluated with the same samples. Sensory evaluation was carried out after approval of the IRB (Institutional Review Board; KHGIRB-20-397).

### 2.5. Statistical Analysis

Each experiment was triplicated, and 3 sub-replicates were used in each experiment. The experimental results of all tests are expressed as the mean ± standard deviation. One-way ANOVA was used to verify the significant difference, and Duncan’s multiple range test was applied using SAS software (version 3.2, SAS Institute, Inc., Cary, NC, USA) at a 95% level. Correlations between experimental results were investigated using Pearson’s correlation coefficients.

## 3. Results and Discussion

### 3.1. Bending Test of Cooked Rice Noodles

Failure stress (σ) data of noodle strands obtained from the bending test are shown in Table 2. In the case of the R1 sample at 80 °C, the failure stress and modulus of elasticity were not obtained because the center of the noodle strand was not fully rehydrated (Table 2 and Table 3). In general, the failure stress of rice noodles containing more than 60% rice content decreased as both cooking temperature and time increased from 80 to 90 °C and from 3 to 4 min, respectively. On the other hand, cooked noodles with lower rice content (R5–R9) did not show any tendency between failure stress and cooking conditions. In R2, noodles cooked at 80 °C for 3 min had the highest failure stress (1.28 kPa). However, as cooking time increased to 4 min, failure stress decreased greatly to 0.48 kPa, and the lowest failure stress, 0.13 kPa, was found for the same noodle cooked at 90 °C for 4 min. R7 with 43% rice content did not appear to be significantly affected by cooking conditions. Overall, rice noodles cooked at 90 °C for 4 min showed the lowest failure stress due to the softening of the tissue during cooking.

The bending test of rice noodles determined their individual failure characteristics. Modulus of elasticity (E) data for noodle strands obtained from the bending test are shown in Table 2. Mostly, the modulus of elasticities of rice noodles cooked at 80 °C revealed higher values than those of rice noodles cooked at 90 °C with some exception. Only two noodles (R2 and R7) showed a decrease in modulus of elasticity with increasing cooking time at 80 °C. But, at 90 °C, the modulus of elasticity of all noodles decreased with increasing cooking time, except R5. Similar to failure stress, the modulus of elasticity of noodles at 90 °C for 4 min revealed the lowest values possibly due to the softening of the tissue. It is interesting to note that the modulus of elasticity of the noodles decreased with an increasing cooking time at 80 °C, and the opposite occurred at 90 °C. This is not consistent with the results of the failure stress obtained during the same bending test, indicating that failure stress and modulus of elasticity are not correlated with each other.

### 3.2. Stress–Relaxation Test of Cooked Rice Noodles

The instantaneous stress, equilibrium stress, elastic modulus (*E*_1_ and *E*_3_), and viscous modulus (*η*_3_) of each sample are shown in Figure 3 as a result of stress–relaxation analysis. Sample R9 showed the highest instantaneous stress, which is consistent with failure stress in the bending test. Sample R4 presented the lowest equilibrium stress, although there were no significant differences between the samples. Using the three-element model, the initial elastic modulus *E*_1_ corresponded to instantaneous stress and was highest for sample R9. In the case of *E*_3_, the late modulus of elasticity, the lowest value was found for sample R2, which seemed to reflect the equilibrium stress. On the other hand, the viscosity coefficient (*η*_3_), an index to distinguish the characteristics of samples, was largest for sample R8 and lowest for R4.

It was difficult to obtain relaxation data due to the spaces within the noodle bundle. If the noodle strands had few empty spaces between them, the deviation would have been smaller [24]. Therefore, the bending test, which was not affected by space, appeared to be more suitable for measuring the rheological properties of the noodle strands.

### 3.3. Sensory Evaluation of Cooked Rice Noodles

The results of the sensory evaluation of the noodles are shown in Table 4. Samples R1, R7, and R8 received higher scores in hardness, indicating a harder texture of the noodle. The hardness of cooked noodles is influenced by the raw materials, the preparation process, and cooking time [25]. Compared to wheat noodles, rice noodles are characterized by stickiness [4]. The highest stickiness was observed for sample R6, whereas R3 had the lowest stickiness. However, the texture of the noodles did not appear to have a decisive impact on overall acceptance. It was difficult to establish a relationship among texture, process conditions, and rice content in this study.

### 3.4. Correlations between the Mechanical Model Parameters and Sensory Evaluation

Correlations between the mechanical parameters obtained from the bending test, stress–relaxation test, and sensory evaluation are shown in Table 5. As expected, failure stress was significantly correlated with modulus of elasticity (*p* < 0.001). In addition, there was a significant positive correlation between failure stress and instantaneous stress (*p* < 0.001), and equilibrium stress had a significant correlation with *E*_3_ (*p* < 0.001). The mechanical parameters and the results of the sensory texture showed relatively weak correlations. In particular, hardness had a positive correlation with stickiness. Sensory stickiness correlated with the modulus of elasticity in the bending test. Both *E*_3_ and *η*_3_ were correlated with overall preference, showing that a higher *E*_3_ and *η*_3_ reflected higher preference. Because it is difficult to measure the mechanical properties of noodles and to create an index of the sensory score because of the noodles’ thin and irregular appearance [26], it is important to set the instrumental tool to measure the mechanical properties of noodles. Currently, the bending test is the most appropriate method to obtain the mechanical properties of noodles.

No strong correlation was observed between mechanical variables and rice content as well as diameter of rice noodles as shown in Table 5 and Table 6. *η*_3_ had a significantly negative correlation with rice content. Due to the limited information on commercially available rice noodles, it was not possible to consider the varieties of rice, how the noodles were made, and the use of additives that could affect the quality characteristics. Since the mechanical parameters of rice noodles did not show a strong correlation with sensory parameters, it is necessary to study other factors such as additives or the size of rice noodles.

## 4. Conclusions

The rheological properties of a strand of various Korean commercial rice noodles were investigated by mechanical tests. In the stress–relaxation test, instantaneous stress and equilibrium stress showed various measurements for each sample. As a result of the bending test and stress–relaxation, these instrumental variables demonstrated differences in the rheological properties of rice noodles in strands and bundles. We also investigated correlations between the quality characteristics of noodles, and correlations between the parameters of the bending and stress–relaxation tests were confirmed. Therefore, when measuring the properties of noodles, it is necessary to classify how each parameter affects the actual sensory characteristic. No strong correlation was observed between mechanical variables and rice content. In this study, commercially available rice noodles were used as samples, and it seems that the results represent the effects of additives rather than those of the rice. Since the mechanical parameters of the texture of rice noodles did not show a strong correlation with the sensory tests, it is necessary to study factors other than the organizational characteristics that influence the preference of Korean rice noodles.

## Figures and Tables

**Figure 1 foods-10-02885-f001:**
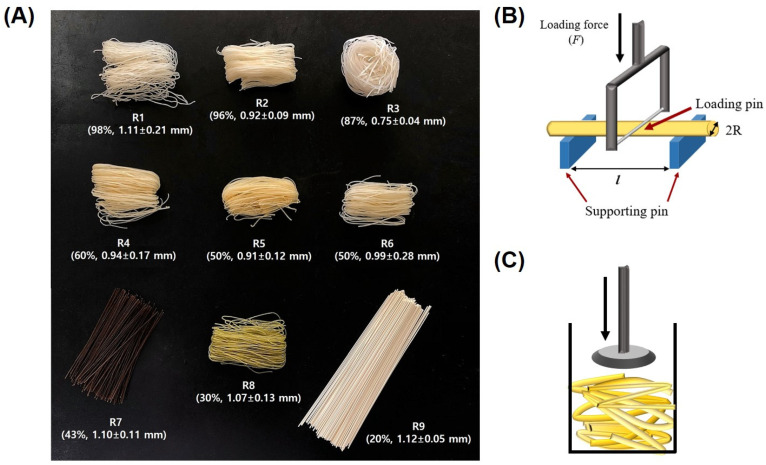
Appearance (**A**) and schematic images of the bending test (**B**) and stress–relaxation test (**C**) of commercial rice noodles.

**Figure 2 foods-10-02885-f002:**
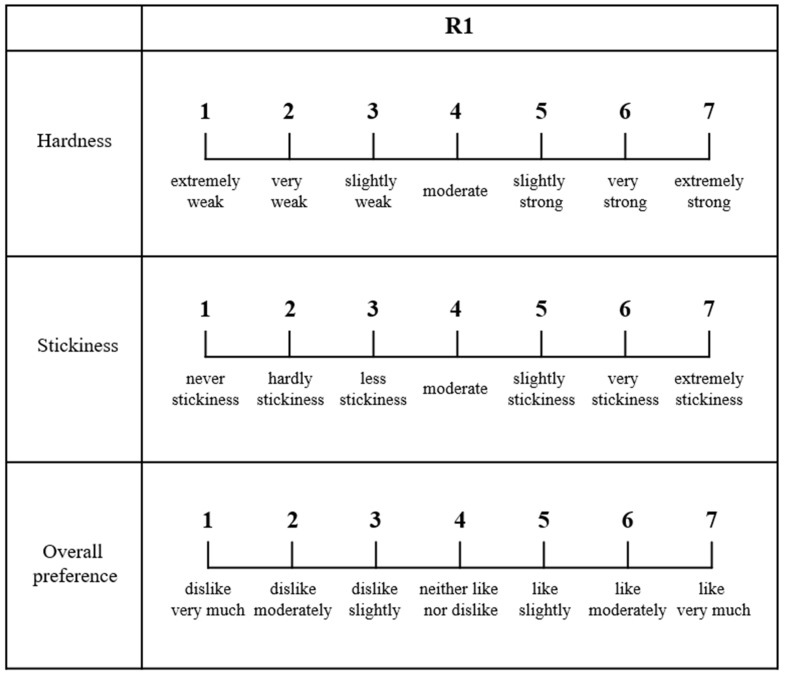
Sensory evaluation form for commercial rice noodles in Korea.

**Figure 3 foods-10-02885-f003:**
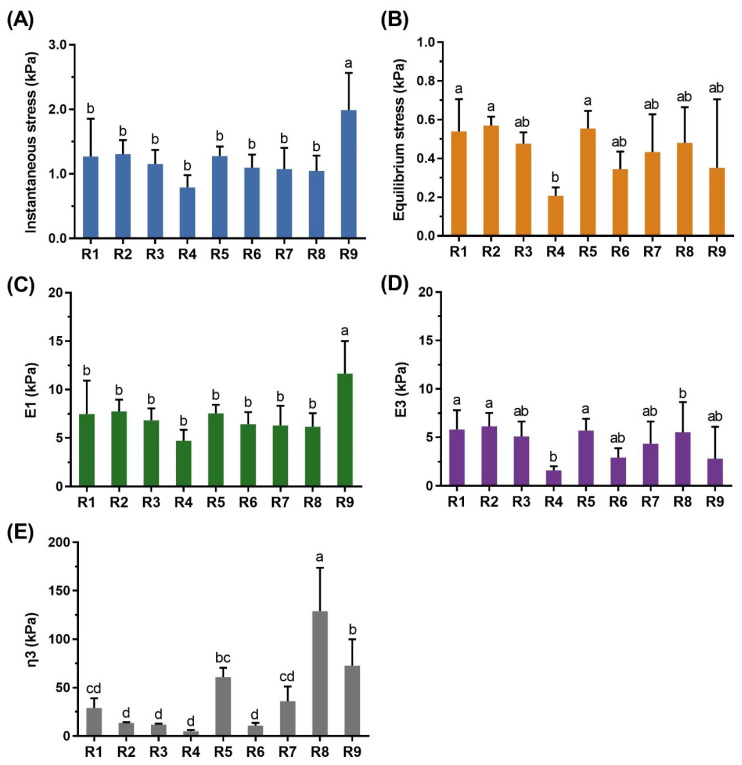
Stress–relaxation test results of cooked rice noodle bundles: instantaneous stress (**A**), equilibrium stress (**B**), *E*_1_ (**C**), *E*_3_ (**D**), and *η*_3_ (**E**). The same letters on the bar in each graph are not significantly different (*p* < 0.05).

**Table 1 foods-10-02885-t001:** Detailed ingredients in rice noodle samples.

	Ingredient
R1	Rice flour (98%, Korea), vegetable oil, and refined salt
R2	Rice flour (96%, Korea), potato starch, modified tapioca starch, alginic acid, and refined salt
R3	Rice flour (87%, Vietnam), modified starch, white sugar, refined salt, and guar gum
R4	Rice four (60%), wheat flour, wheat starch, tapioca starch, and refined salt
R5	Rice flour (50%, Korea) wheat flour (34.4%), wheat starch, and refined salt
R6	Rice flour (50%, Korea), wheat flour, corn starch, modified starch, dextrin, guar gum, refined salt, and alkaline reagent
R7	Rice flour (43%, Korea), wheat flour, seaweed concentrated (*Hizikia fusiforme*, 16.3%), starch, and refined salt
R8	Rice flour (30%, Korea), wheat flour (53.8%, Korea), lotus leaf powder (2%), starch, and refined salt
R9	Rice flour (20%, Korea), wheat flour (78%, Korea), and refined salt (2%)

**Table 2 foods-10-02885-t002:** Diameter of rice noodles before cooking and the failure stress (σ) of cooked rice noodle strands.

	Diameter (mm)	Failure Stress (σ, kPa)			
		80 °C		90 °C	
		3 min	4 min	3 min	4 min
R1	1.11 ± 0.21	N.d ***	N.d	0.47 ± 0.12 ^B,a^	0.38 ± 0.12 ^B,a^
R2	0.92 ± 0.09	1.28 ± 0.21 ^A,^*^,a,^**	0.48 ± 0.05 ^C,b^	0.54 ± 0.18 ^B,b^	0.13 ± 0.01 ^C,c^
R3	0.75 ± 0.04	1.03 ± 0.23 ^B,a^	0.76 ± 0.28 ^B,b^	0.38 ± 0.09 ^BC,c^	0.30 ± 0.04 ^B,c^
R4	0.94 ± 0.17	0.70 ± 0.29 ^C,a^	0.43 ± 0.13 ^C,b^	0.23 ± 0.04 ^D,c^	0.14 ± 0.05 ^C,d^
R5	0.91 ± 0.12	0.43 ± 0.02 ^D,a^	0.36 ± 0.05 ^D,b^	0.20 ± 0.02 ^D,c^	0.30 ± 0.13 ^B,b^
R6	0.99 ± 0.28	0.42 ± 0.24 ^D,a^	0.34 ± 0.22 ^D,b^	0.20 ± 0.04 ^D,c^	0.20 ± 0.07 ^C,c^
R7	1.10 ± 0.11	0.26 ± 0.08 ^E,a,b^	0.19 ± 0.04 ^E,b^	0.33 ± 0.17 ^C,a^	0.37 ± 0.14 ^B,a^
R8	1.07 ± 0.13	0.43 ± 0.19 ^D,a^	0.53 ± 0.13 ^C,a^	0.43 ± 0.26 ^B,a^	0.18 ± 0.02 ^C,b^
R9	1.12 ± 0.05	1.01 ± 0.11 ^B,b^	2.48 ± 1.25 ^A,a^	1.15 ± 0.17 ^A,b^	1.20 ± 0.53 ^A,b^

* The same capital letters in the same column are not significantly different (*p* < 0.05). ** The same lowercase letters in the same row are not significantly different (*p* < 0.05). *** N.d = no data.

**Table 3 foods-10-02885-t003:** Modulus of elasticity (E) of the rice noodle strands.

Sample	Modulus of Elasticity (mPa)			
	80 °C		90 °C	
	3 min	4 min	3 min	4 min
R1	N.d	N.d ***	525 ± 53 ^A,a^	419 ± 64 ^A,b^
R2	599 ± 32 ^A,^*^,a,^**	293 ± 28 ^B,C,c^	356 ± 33 ^C,b^	160 ± 32 ^D,d^
R3	447 ± 11 ^B,a^	467 ± 77 ^B,a^	317 ± 27 ^C,b^	253 ± 24 ^B,c^
R4	322 ± 32 ^C,b^	402 ± 92 ^B,b,a^	267 ± 45 ^D,c^	179 ± 10 ^D,d^
R5	317 ± 42 ^C,a,b^	347 ± 97 ^B,C,a^	229 ± 29 ^D,c^	286 ± 75 ^B,b,c^
R6	283 ± 74 ^D,a^	278 ± 35 ^C,a^	238 ± 18 ^D,a,b^	218 ± 37 ^C,b^
R7	363 ± 76 ^C,a^	217 ± 18 ^D,b^	322 ± 91 ^C,a^	316 ± 70 ^B,a^
R8	303 ± 71 ^C,b^	370 ± 40 ^B,C,a^	263 ± 53 ^D,b,c^	200 ± 19 ^C,c^
R9	363 ± 14 ^C,c^	673 ± 326 ^A,a^	424 ± 48 ^B,b^	400 ± 170 ^A,b,c^

* The same capital letters in the same column are not significantly different (*p* < 0.05). ** The same lowercase letters in the same row are not significantly different (*p* < 0.05). *** N.d = no data.

**Table 4 foods-10-02885-t004:** Sensory evaluation of cooked rice noodles.

Sample	Hardness	Stickiness	Overall Preference
R1	5.67 ± 0.98 ^a,^*	3.40 ± 1.35 ^c^	4.33 ± 1.45 ^a^
R2	3.60 ± 1.18 ^c^	4.53 ± 1.77 ^a,b^	4.47 ± 1.13 ^a^
R3	3.80 ± 1.32 ^c^	3.27 ± 1.10 ^c^	3.93 ± 1.39 ^a,b^
R4	4.73 ± 0.96 ^b^	4.20 ± 1.21 ^b^	4.80 ± 0.94 ^a^
R5	4.40 ± 1.80 ^b^	4.13 ± 1.25 ^b^	4.00 ± 1.13 ^a,b^
R6	3.60 ± 1.45 ^c^	5.20 ± 1.42 ^a^	4.13 ± 1.55 ^a,b^
R7	5.47 ± 1.68 ^a^	4.00 ± 1.25 ^b^	3.53 ± 1.60 ^b^
R8	5.00 ± 1.07 ^a,b^	3.67 ± 1.45 ^c^	3.47 ± 1.51 ^b^
R9	3.87 ± 1.51 ^c^	3.67 ± 1.29 ^c^	4.47 ± 1.64 ^a^

* The same letters in the same column are not significantly different (*p* < 0.05).

**Table 5 foods-10-02885-t005:** Pearson correlation coefficients between mechanical and sensory test parameters.

	Diameter	FS	ME	IS	ES	*E* _3_	*η* _3_	SH	SS	SO
Diameter	1									
FS	0.29	1								
ME	0.33	0.60 ***	1							
IS	0.07	0.68 ***	0.30	1						
ES	0.23	0.04	0.15	0.36	1					
*E* _3_	0.08	−0.03	0.13	0.19	0.95 ***	1				
*η* _3_	0.36	0.34	−0.06	0.33	0.35	0.38 *	1			
SH	0.61 ***	0.11	0.40	0.33	0.09	0.16	0.26	1		
SS	0.48 *	0.35	0.52 **	0.23	0.25	0.30	0.34	0.43 *	1	
SO	0.44 *	0.17	0.01	0.19	0.39 *	0.44 *	0.54 **	0.31	0.20	1

FS = failure stress (kPa); ME = modulus of elasticity (mPa); IS = instantaneous stress (kPa); ES = equilibrium stress (kPa); SH = sensory hardness; SS = sensory stickiness; SO = sensory overall acceptance. * *p* < 0.05, ** *p* < 0.01, and *** *p* < 0.001

**Table 6 foods-10-02885-t006:** Pearson correlation coefficients between characteristic parameters and the rice content of noodles.

	FS	ME	IS	ES	*E* _3_	*η* _3_	SH	SS	SO
Rice content	−0.27	0.38 *	−0.25	0.43 *	0.45 *	−0.65 **	0.02	0.10	0.32

FS = failure stress (kPa); ME = modulus of elasticity (mPa); IS = instantaneous stress (kPa); ES = equilibrium stress (kPa); SH = sensory hardness; SS = sensory stickiness; SO = sensory overall acceptance. ** p <* 0.05 and ** *p <* 0.01.

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
