# Peer review of "Instrumental and Sensory Characteristics of Commercial Korean Rice Noodles"

_foods, 2021, doi:10.3390/foods10112885_

Round 1

Reviewer 1 Report

This is an interesting article but it fails in a number of  places. I wonder wether you are using the best ways to present your data.  In several places you offer tables and with mean and standard deviations, but figure 2 is a set of bar charts - is this just for variety or is a bar chart better than a table for this kind of data?

You do provide images of the noodles (Figure 1), but how do they differ?  Perhaps a table outlining those differences would help the paper for example in the discussion you talk about rice content, this could form a comparison as could average diameter and the presence of other ingredients, etc...

I assume N.d in tables 1 and 2 means "no data" if so can you say so, if not then please explain.  Assuming the former, why is there no data?

lines 164 and 169, should this read "the same column"?

You say the samples were assessed by a trained taste panel, please explain how they were selected and trained?

Author Response

Response to Reviewer 1

This is an interesting article but it fails in a number of places.

  1. I wonder whether you are using the best ways to present your data. In several places you offer tables and with mean and standard deviations, but figure 2 is a set of bar charts - is this just for variety or is a bar chart better than a table for this kind of data?

Ans) Thank you for your suggestion. We described those results as figure because the figure would be more intuitive than the table and it would be helpful to catch the overall tendency of the noodle samples with different rice flour content readily.

  1. You do provide images of the noodles (Figure 1), but how do they differ?  Perhaps a table outlining those differences would help the paper for example in the discussion you talk about rice content, this could form a comparison as could average diameter and the presence of other ingredients, etc...

Ans) Thank you for your suggestion. We added the detailed composition of the rice noodle sample in supplementary data (Table S1) as you suggested. Please see the attached supplementary data.

  1. I assume N.d in tables 1 and 2 means "no data" if so can you say so, if not then please explain. Assuming the former, why is there no data?

Ans) Thank you for your suggestion. In the case of the R1 sample at 80℃, failure stress and modulus of elasticity were not obtained because the center of the noodle strand was not fully rehydrated. We rewrote them as you suggested. Please see lines 143-144 in the revised manuscript.

  1. lines 164 and 169, should this read "the same column"?

Ans) Thank you for your suggestion. We carried out statistical analysis between the data obtained from several rehydration conditions within the same sample. Therefore, the notation 'same row' on lines 168 and 172 in the revised manuscript is appropriate.

  1. You say the samples were assessed by a trained taste panel, please explain how they were selected and trained?

Ans) Thank you for your suggestion. We selected the following people as the panel for a sensory test; 1) people with taste evaluation and expression skills normally 2) people who are not allergic to certain foods and 3) people who are not color- and taste-blindness. Also, we performed the following training for the panel before the sensory evaluation; 1) explanation of sensory evaluation method 2) tasting method (spoon, cup, drink, etc), and 3) a specific description of the question, term, scale in the evaluation form.

Reviewer 2 Report

Dear authors, 

After reading the manuscript : " Instrumental and sensory characteristics of  commercial Korean rice noodles" , I realized that the manuscript showed in some parts the scientific rigour wanted, but in other parts I have missed it. The authors have presented critical evaluation only in some paragraphs, besides the title and objective need some improvement. That's why I have written some suggestions in an attempt to improve these parts.

L.47- What would be a representative product? In all countries ? is it ? For which group ? economically ? More details, please.

L.52- modification? More details, please.

L.61-62- I think we have to be a little careful with this sentence. We have great results in sensory tests and excellent researches.
The cited paper is from 2001 and we need to consider that sensory analysis had a great advance in the last 20 years. Besides, the paper that you mention this sentence is only about descriptive analysis, and we would have other types and tests.

L.79- Exactly. It reinforces what I think. The two evaluations together are very important.

L. 84- Explain the difference to the industrial, why does it deserve this highlight ?

L.84- 86- I wonder if "technological" wouldn't be more adequate than "mechanical".

L.88- "and rice content"- I did not understand this part. Why only at the end ?

L.95- Figure 1 looks very beautiful, but I missed a table with the amounts of ingredients of the noodles, it would even emphasize more the amount of rice, which I think is important in the paper.

L.125-  For the sensory test, did you submit the project to an ethics committee? Please insert the approval protocol.

L.126- We need more details about sensory testing. What is the profile of the assessors?  Are they used to consuming noodles ? How was the training conducted and what were the conditions ? What axes are the scales anchored on ? We need more infirmation in this section, please.

L. 127- 128- hardness and stickiness - Why only these attributes were evaluated?

L.197- We need to revisit the seven-point scale in the discussion, therefore it is important to know all the points of the scale.

L.199- Overall Preference  was not mentioned in Material and methods.

L. 233- Standardize the nomenclature : Mechanical, rheological, physical

L.235- Cooking conditions- I don't know if we can bring "Cooking conditions" into the results and conclusion. It didn't seem to me to be an objective of the paper and it really didn't appear before.

Some things were not clear to me and I would appreciate if the authors could improve paper and clarify:
L.59- I did not understand why address noodles with chickpea here ?
L.45- Why didn't you advance in the proposal of gluten-free as a differential of the paper, you only mentioned it in the introduction and it got confusing.
What was the standard treatment ?

L.266- Some references are not within the guidelines of the journal;

Author Response

Response to Reviewer 2

After reading the manuscript : " Instrumental and sensory characteristics of commercial Korean rice noodles" , I realized that the manuscript showed in some parts the scientific rigour wanted, but in other parts I have missed it. The authors have presented critical evaluation only in some paragraphs, besides the title and objective need some improvement. That's why I have written some suggestions in an attempt to improve these parts.

L.47- What would be a representative product? In all countries ? is it ? For which group ? economically ? More details, please.

Ans) Thank you for your suggestion. We rewrite the content in more detail. Please see lines 46-48 in the revised manuscript.

L.52- modification? More details, please.

Ans) Thank you for your suggestion. We add the detailed information. Please see line 52 in the revised manuscript.

L.61-62- I think we have to be a little careful with this sentence. We have great results in sensory tests and excellent researches.

Ans) Thank you for your suggestion. We rewrote this sentence as you suggested. Please see lines 60-63 in the revised manuscript.

The cited paper is from 2001 and we need to consider that sensory analysis had a great advance in the last 20 years. Besides, the paper that you mention in this sentence is only about descriptive analysis, and we would have other types and tests.

Ans) Thank you for your suggestion. We rewrote this part as you suggested. Please see lines 54-63 in the revised manuscript.

L.79- Exactly. It reinforces what I think. The two evaluations together are very important.

Ans) Thank you for your suggestion. We also have the same opinion.

  1. 84- Explain the difference to the industrial, why does it deserve this highlight ?

Ans) Thank you for your suggestion. We mentioned “the industrial formulation” in the previous manuscript because most Korean rice noodles are industrially produced by extrusion. We deleted “the industrial formulation” as you suggested. Please see lines 83-84 in the revised manuscript.

L.84- 86- I wonder if "technological" wouldn't be more adequate than "mechanical".

Ans) Thank you for your suggestion. We changed it as you suggested. Please see line 85 in the revised manuscript.

L.88- "and rice content"- I did not understand this part. Why only at the end ?

Ans) Thank you for your suggestion. We used several commercial rice noodles for this study and they have different rice flour content among samples. Since this difference in rice flour content was expected to be one of the factors that affect the mechanical and sensory properties, a correlation for rice flour content was also carried out. However, it may confuse the objective of this work, we delete “the rice flour content” as you suggested. Please see lines 88-89 in the revised manuscript.

L.95- Figure 1 looks very beautiful, but I missed a table with the amounts of ingredients of the noodles, it would even emphasize more the amount of rice, which I think is important in the paper.

Ans) Thank you for your suggestion. We added the detailed composition of the rice noodle samples in supplementary data (Table S1). Please see the attached supplementary data.

L.125-  For the sensory test, did you submit the project to an ethics committee? Please insert the approval protocol.

Ans) Thank you for your suggestion. We added the IRB number (KHGIRB-20-397) as you suggested. Please see the line 131 in the revised manuscript.

L.126- We need more details about sensory testing. What is the profile of the assessors?  Are they used to consuming noodles? How was the training conducted and what were the conditions? What axes are the scales anchored on? We need more information in this section, please.

Ans) Thank you for your suggestion. We selected the following people as the panel for a sensory test; 1) people with taste evaluation and expression skills normally 2) people who are not allergic to certain foods and 3) people who are not color- and taste-blindness. Also, we performed the following training for the panel before the sensory evaluation; 1) explanation of sensory evaluation method 2) tasting method (spoon, cup, drink, etc), and 3) a specific description of the question, term, scale in the evaluation form. Additionally, we add the sensory evaluation form in supplementary data (Figure S1). Please see the attached supplementary data.

  1. 127- 128- hardness and stickiness - Why only these attributes were evaluated?

Ans) Thank you for your suggestion. We considered that the hardness and stickiness of the sensory evaluation are the indexes that can be expressed the correspondence with the parameters obtained from the mechanical analysis (bending test and stress relaxation test).

L.197- We need to revisit the seven-point scale in the discussion, therefore it is important to know all the points of the scale.

Ans) Thank you for your suggestion. As mentioned earlier, we add the sensory evaluation form in supplementary data (Figure S1). Please see the attached supplementary data.

L.199- Overall Preference was not mentioned in Material and methods.

Ans) Thank you for your suggestion. We added it. Please see line 130 in the revised manuscript.

  1. 233- Standardize the nomenclature : Mechanical, rheological, physical

Ans) Thank you for your suggestion. We checked the manuscript and standardize the nomenclature (physical → rheological) as you suggested. Please see line 239 in the revised manuscript.

L.235- Cooking conditions- I don't know if we can bring "Cooking conditions" into the results and conclusion. It didn't seem to me to be an objective of the paper and it really didn't appear before.

Ans) Thank you for your suggestion. We deleted this part. Please see section 4 in the revised manuscript.

Some things were not clear to me and I would appreciate if the authors could improve paper and clarify:

L.59- I did not understand why address noodles with chickpea here ?

Ans) Thank you for your suggestion. We did not intend to emphasize ‘chickpea’, but rather as reporting the previous study focused on physicochemical and sensory properties of rice noodles with additives in general.

L.45- Why didn't you advance in the proposal of gluten-free as a differential of the paper, you only mentioned it in the introduction and it got confusing. What was the standard treatment ?

Ans) Thank you for your suggestion. Gluten-free is the general terminology and advantage of rice grains. In the introduction, we tried to explain the general advantages of rice grains. There are no standard treatments regarding gluten-free products in this study.

L.266- Some references are not within the guidelines of the journal;

Ans) Thank you for your suggestion. We rewrote the references according to guidelines. Please see the reference section in the revised manuscript.

Round 2

Reviewer 1 Report

I should have noticed in the first submission that you have nothing about ethics and consent in the sensory testing section. This must be included.

Why do you include supplementary information for essential material?  In response to my previous review you have included information in a supplementary section - in reality most readers will not look at the supplementary information and therefor it is lost.  Please incorporate these changes into the main body of your paper!!

You kindly provided information on how the sensory assessors were selected and trained.  BUT you have not incorporated this into the paper. You MUST add this information in the material and methods section!!

Please be aware that "preference" is not a sensory test. Preference tells us about the assessors and not about the products e.g. the South east asian fruit Durian is hated by most europeans, yet highly desirable in many asian countries.

Please can you double check with your statistician about the footnote to tables 1 & 2.  From what you say, it is about rows - which means you are comparing cooking times/temperatures, yet the discussion which follows compares noodle types. To my mind you are comparing the different noodles and not the cooking procedures.  This needs to be clear.  Similarly, in Figure 2, there are lower case letters above the bars which need explaining in the legend.

Line 198 you use the word "significant", do you mean this in a statistical sense? If so, please give the confidence level, if not - please choose another word.

Table 3, "overall preference" column, can you have "ab" of "b" does not occur on its own? I do not think so.  If all the samples are labelled "a" and none are labelled "b" then you cannot have "ab". Similarly the "hardness" column does not have "c"or "b" on their own, so how can you have "cd" or "abc" or "bcd" etc...?  ...and your stickiness column has not got "b" on its own.

Author Response

Response to Reviewer 1

  1. I should have noticed in the first submission that you have nothing about ethics and consent in the sensory testing section. This must be included.

Ans) Thank you for your suggestion. We added IRB number and information about this approval. Please see lines 137-139 in the revised manuscript.

  1. Why do you include supplementary information for essential material?  In response to my previous review you have included information in a supplementary section - in reality most readers will not look at the supplementary information and therefor it is lost.  Please incorporate these changes into the main body of your paper!!

Ans) Thank you for your suggestion. We added the information of essential material into the body text. Please see Table 1 in the revised manuscript.

  1. You kindly provided information on how the sensory assessors were selected and trained.  BUT you have not incorporated this into the paper. You MUST add this information in the material and methods section!!

Ans) Thank you for your suggestion. We added the information on the sensory test in more detail. Please see the lines 131-136 in the revised manuscript.

  1. Please be aware that "preference" is not a sensory test. Preference tells us about the assessors and not about the products e.g. the South east asian fruit Durian is hated by most europeans, yet highly desirable in many asian countries.

Ans) Thank you for your suggestion. Many studies carried out the overall preference also investigated as one of the test parameters in sensory evaluation. However, we would be aware and keep this in mind as you suggested.

  1. Please can you double check with your statistician about the footnote to tables 1 & 2.  From what you say, it is about rows - which means you are comparing cooking times/temperatures, yet the discussion which follows compares noodle types. To my mind you are comparing the different noodles and not the cooking procedures.  This needs to be clear.  Similarly, in Figure 2, there are lower case letters above the bars which need explaining in the legend.

Ans) Thank you for your suggestion. We carried out additional statistical analysis for comparing the significance among samples. Also, we explained the lower case on bar in Figure 3 as footnote. Please see the Table 2 and 3, and Figure 3 in the revised manuscript.

  1. Line 198 you use the word "significant", do you mean this in a statistical sense? If so, please give the confidence level, if not - please choose another word.

Ans) Thank you for your suggestion. We replaced the word “significant” to another appropriate word. Please see line 212 in the revised manuscript.

  1. Table 3, "overall preference" column, can you have "ab" of "b" does not occur on its own? I do not think so.  If all the samples are labelled "a" and none are labelled "b" then you cannot have "ab". Similarly the "hardness" column does not have "c"or "b" on their own, so how can you have "cd" or "abc" or "bcd" etc...?  ...and your stickiness column has not got "b" on its own.

Ans) Thank you for your suggestion. We performed the statistical analysis again and the significance was rewritten. Please see Table 4 in the revised manuscript.

Reviewer 2 Report

After another evaluation of the manuscript, I see a great improvement in the quality of the paper. The authors have accepted almost all of my requests. They added more authors to better substantiate the methodology and improved tables.  But, I think this part below would be very important and should be in the paper, and so is the table with the formulations, and not in the supplementary document.

"We selected the following people as the panel for a sensory test; 1) people with taste evaluation and expression skills normally 2) people who are not allergic to certain foods and 3) people who are not color- and taste-blindness. Also, we performed the following training for the panel before the sensory evaluation; 1) explanation of sensory evaluation method 2) tasting method (spoon, cup, drink, etc), and 3) a specific description of the question, term, scale in the evaluation form. Additionally, we add the sensory evaluation form in supplementary data (Figure S1). Please see the attached supplementary data."

Author Response

Response to Reviewer 2

  1. After another evaluation of the manuscript, I see a great improvement in the quality of the paper. The authors have accepted almost all of my requests. They added more authors to better substantiate the methodology and improved tables.  But, I think this part below would be very important and should be in the paper, and so is the table with the formulations, and not in the supplementary document.

We selected the following people as the panel for a sensory test; 1) people with taste evaluation and expression skills normally 2) people who are not allergic to certain foods and 3) people who are not color- and taste-blindness. Also, we performed the following training for the panel before the sensory evaluation; 1) explanation of sensory evaluation method 2) tasting method (spoon, cup, drink, etc), and 3) a specific description of the question, term, scale in the evaluation form. Additionally, we add the sensory evaluation form in supplementary data (Figure S1). Please see the attached supplementary data."

Ans) Thank you for your suggestion. We added these sentences and evaluation paper that you mentioned. Please see the line 130-135 and Figure 2 in the revised manuscript.